# Efficacy of Six Different SARS-CoV-2 Vaccines during a Six-Month Follow-Up and Five COVID-19 Waves in Brazil and Mexico

**DOI:** 10.3390/vaccines11040842

**Published:** 2023-04-14

**Authors:** Maria Elena Romero-Ibarguengoitia, Diego Rivera-Salinas, Riccardo Sarti, Riccardo Levi, Maximiliano Mollura, Arnulfo Garza-Silva, Andrea Rivera-Cavazos, Yodira Guadalupe Hernández-Ruíz, Irene Antonieta Barco-Flores, Arnulfo González-Cantú, Miguel Ángel Sanz-Sánchez, Milton Henriques Guimarães Júnior, Chiara Pozzi, Riccardo Barbieri, Devany Paola Morales-Rodriguez, Mauro Martins Texeira, Maria Rescigno

**Affiliations:** 1Research Department, Hospital Clínica Nova de Monterrey, San Nicolas de los Garza 66450, Nuevo Leon, Mexico; 2Vicerrectoría de Ciencias de la Salud, Escuela de Medicina, Universidad de Monterrey, San Pedro Garza García 66238, Nuevo Leon, Mexico; 3Department of Biomedical Sciences, Humanitas University, Via Rita Levi Montalcini 4, Pieve Emanuele, 20072 Milan, Italy; 4IRCCS Humanitas Research Hospital, Via Manzoni 56, Rozzano, 20089 Milan, Italy; 5Politecnico di Milano, Department of Electronic, Information and Bioengineering, 20133 Milan, Italy; 6Research Department Fundação São Francisco Xavier, Bairro das Águas 35160-158, Minas Gerais, Brazil; 7Biochemistry and Immunology Department, ICB, Universidade Federal de Minas Gerais, Belo Horizonte 31270-901, Minas Gerais, Brazil

**Keywords:** vaccination, SARS-CoV-2, COVID-19, coronavirus, immunization, antibodies

## Abstract

Comparisons among the different vaccines against SARS-CoV-2 are important to understand which type of vaccine provides more protection. This study aimed to evaluate the real-life efficacy through symptomatic infection and the humoral response of six different vaccines against SARS-CoV-2—BNT162b2, mRNA-1273, ChAdOx1-S, CoronaVac, Ad26.COV2, and Ad5-nCoV. This multicentric observational longitudinal study involved hospitals from Mexico and Brazil in which volunteers who received complete vaccination schemes were followed for 210 days after the last dose. SARS-CoV-2 Spike 1–2 IgG levels were taken before receiving the first vaccine, 21 days after each dose, and the last sample at six months (+/−1 month) after the last dose. A total of 1132 individuals exposed to five COVID-19 waves were included. All vaccines induced humoral responses, and mRNA vaccines had the highest antibody levels during follow-up. At six months, there was a decline in the SARS-CoV-2 Spike 1–2 IgG antibody titers of 69.5% and 36.4% in subjects with negative and positive history of infection respectively. Infection before vaccination and after complete vaccination scheme correlated with higher antibody titers. The predictors of infection were vaccination with CoronaVac compared to BNT162b2 and ChAdOx1-S. In the presence of comorbidities such as diabetes, rheumatoid arthritis, or dyslipidemia, CoronaVac lowered the risk of infection.

## 1. Introduction

Vaccination against SARS-CoV-2 is a crucial worldwide strategy for the control of the COVID-19 pandemic. Immunization has proven to be safe and effective in reducing severe cases, hospitalization, and death by COVID-19 [1]. Despite the mutability of the virus and its capacity to genetically evolve into new variants, vaccination is still imperative because of its proven effectiveness in preventing severe disease in variants of concern (VOCs), including the Alpha, Beta, Gamma, and Delta variants [2]. New information is required concerning Omicron VOCs.

By December 2022, there were 201 countries with at least one approved vaccine [3]. Vaccines are most usefully classified according to their mechanism of action. RNA vaccines, like BNT162b2 (BioNTech and Pfizer, New York, NY, USA) and mRNA-1273 (Moderna, Cambridge, MA, USA), use nanoparticle-modified viral antigen-encoding mRNA to induce humoral and cellular immunity against SARS-CoV-2. Viral vector vaccines, such as ChAdOx1-S (AstraZeneca, Cambridge, UK), Ad26.COV2.S (Johnson & Johnson, New Brunswick, NJ, USA), and Ad5-vCoV (CanSino Biologics, Tianjin, China), induce an immune response by inoculating genetically modified viruses that express an antigen of interest mimicking a natural infection. Whole inactivated vaccines as CoronaVac (Sinovac Biotech, Beijing, China) employ cultured inactivated viral particles containing antigens of the pathogen of interest able to induce immune responses [4].

The waning of vaccination’s effectiveness against SARS-CoV-2 has been a concern, and very little is known about protection from inactivated vaccines. Previous studies have shown a decrease in effectiveness around six months after a complete vaccination scheme with BNT162b2, mRNA-1273, ChAdOx1-S, or Ad5-nCoV [5,6]. Regarding humoral response, Khoury and collaborators reported a drop in antibody titers one month after a two-dose vaccination scheme, with a mean antibody titer of 6% of the peak level after four months with BNT162b2 [7]. Favresse and collaborators also concluded that a significant decline in antibody titers is noticeable after three months of vaccination with BNT162b2 [8]. Long-time follow-up of vaccinated subjects is required to understand the effectiveness of available vaccines through different waves and new VOCs of SARS-CoV-2, such as Omicron. Additionally, comparisons among different vaccine types are mandatory to understand which kind of vaccine provides more protection against SARS-CoV-2 infection and other predictors that are related to symptomatic infection. 

Therefore, this study aimed to evaluate the real-life efficacy of six different vaccines against SARS-CoV-2—BNT162b2, mRNA-1273, ChAdOx1-S, CoronaVac, Ad26.COV2, and Ad5-nCoV. Efficacy was measured through symptomatic infection and humoral response. Individuals were followed for 210 days and were exposed to various SARS-CoV-2 VOCs, including Omicron, during this study. To assess the humoral immune response, SARS-CoV-2 Spike 1–2 IgG levels were measured in vaccinated individuals following a shared study design and with the same technique across all countries participating in the study.

## 2. Materials and Methods

This was a multicentric observational longitudinal study that involved two hospital centers (Hospital Clinica Nova and Fundacion San Francisco Xavier) from two different countries (Mexico and Brazil) in which volunteers who received complete schemes of approved vaccines (BNT162b2, mRNA-1273, CoronaVac, ChAdOx1-S, Ad26.COV2, or Ad5-nCoV) were followed for 210 days after the last dose.

This study was designed following the Strengthening the Reporting of Observational studies in Epidemiology (STROBE) guidelines and approved by each of the local Institutional Review Boards, and conducted as per the Code of Ethics of the World Medical Association (Declaration of Helsinki) for experiments involving humans [9]. The inclusion criteria were volunteers of any age, both genders, who consented to participate, planned on completing the vaccination scheme, and agreed to be followed through the study’s duration. The exclusion criteria were having received any SARS-CoV-2 vaccination prior to the study’s development or receiving an additional dose in the following six (+/− one) months after completing the scheme. 

The availability of vaccines was defined by the Health System of Brazil or Mexico at the time individuals were enrolled. The date the subject received their first and second dose could vary depending on the age group. Subjects received the doses during 2021–2022 and were exposed to different waves that included different variants, such as Alpha, Beta, Gamma, Delta, Epsilon, Eta, and Omicron strains. Before receiving the first dose, the research team explained the project and invited the subjects to participate. Those interested in participating were given a consent form which they signed if they agreed. Inclusion and exclusion criteria were applied, and a plasma sample was taken. Serum SARS-CoV-2 Spike 1–2 IgG antibodies were measured. The baseline sample was taken before receiving the first dose of any vaccine (T0); the second (T1) and third (T2) samples were taken 21 days (+/−7 days) after each dose was applied. After six months (+/−1 month), a fourth sample (T3) was taken.

In every sample follow-up, the participants were given a questionnaire where their medical history, vaccination scheme, and SARS-CoV-2 infection history (symptoms and management) before and during the follow-up were obtained. The last questionnaire, applied six months (+/−1 month) after the last dose, recovered information about SARS-CoV-2 infection history after vaccination.

SARS-CoV-2 Spike 1–2 IgG antibodies were measured quantitatively using DiaSorin’s chemiluminescence immunoassay (CLIA). This assay had a sensitivity of 97.4% (95% CI, 86.8–99.5) and a specificity of 98.5% (95 CI, 97.5–99.2). The interpretation of the results was as follows: (1) a negative result for values <12.0 AU/mL; (2) an indeterminate result for 12.0 to 15.0 AU/mL; and (3) a positive result for values >15 AU/mL [10]. This kit has been previously used in multiple studies [11,12,13].

The variables analyzed were sex, age, medical history (i.e., type 2 diabetes mellitus, hypertension, obesity, rheumatoid arthritis, dyslipidemia, cancer, or any other kind of disease), history of confirmed SARS-CoV-2 infection (by nasal swab and PCR or viral protein antigen detection), medical management (ambulatory or hospitalization) and need of supplementary oxygen. Other variables included were the antibody titers previously mentioned: basal, 21–28 days after the first and second dose, and after six months (+/−1 month). Efficacy was measured through symptomatic infection and humoral response at different time points. 

### Statistical Analysis

Researchers assessed the data’s quality control and anonymity. Normality was evaluated through Shapiro–Wilk or Kolmogorov–Smirnoff tests. Descriptive statistics such as mean, standard deviation, median, interquartile range, frequencies, and percentages were computed. The chi-square test was used to compare vaccine groups’ medical history and SARS-CoV-2 infection history. The Kruskal–Wallis test was used for age and body mass index (BMI) differences. Mann–Whitney U and Kruskal–Wallis tests were used to compare antibody titers between vaccine groups. The Friedman test was computed to compare the antibody titers over time. 

An ordinary least square model was used to predict the antibody titers at the 6-month (+/−one month) follow-up, including only subjects with known antibody titers after the 2nd vaccine dose and at the 6-month follow-up. This model included the following covariates: sex, age (standardized), body mass index (standardized), the antibody level after the 2nd dose (standardized), the vaccine type (with BNT162b2 as a reference), any previous SARS-CoV-2 infection (before the 1st dose), any SARS-CoV-2 infection after the 2nd dose, and finally an interaction term between the antibody levels after the 2nd dose and the vaccine administered, to account for the possibly different immune responses induced by the other vaccines (with BNT162b2 as a reference). The target variable, consisting of antibody titers at the 6-month follow-up, was also standardized.

Survival curve analysis and Cox proportional hazard model were used to determine predictor factors for SARS-CoV-2 infection. Since patients received the vaccine scheme at different time points according to local government indications, some patients could finish follow-up before the Omicron wave or after, so an additional variable was created to account for the start date of the Omicron wave. The Cox proportional hazard model was applied by stratifying by this dichotomous variable, which indicated whether a person had been followed until 15 December 2021 (value = 0) or after this cut-off date (value = 1). The model included as covariables: sex (reference: female), age (standardized), body mass index (standardized, then squared to account for its possible non-linear contribution), having been infected with SARS-CoV-2 before the vaccination regimen started, the vaccine type, the antibody level after the 2nd dose (standardized), the presence of at least one important comorbidity among diabetes, arthritis, and dyslipidemia, and finally an interaction term between the vaccine type and the presence of comorbidities—as these appeared to be correlated in a univariate analysis. Due to the small number of persons vaccinated with mRNA-1273, Ad26.COV2, or Ad5-nCoV, those vaccines were excluded from the Cox proportional hazard model. Additionally, we only included persons with no missing values in the model. The event was the first SARS-CoV-2 infection after the 2nd dose. The time-to-event was the days between the 2nd dose and the SARS-CoV-2 infection. Persons with no SARS-CoV-2 infections after the 2nd dose were considered censored cases, with the days elapsed between the 2nd dose and the follow-up date as the time-to-censoring. Since participants were followed for six months (+/−1 month), the end-point for the Cox model was 210 days. 

A *p*-value less than 0.05 was considered statistically significant. Missing random values were analyzed through complete case analysis since missing antibody levels were less than 5%. The statistic programs used were R v. 4.0.3 and Python v. 3.8.3.

## 3. Results

A total of 1132 individuals were included in the multicentric study: 1061 from Mexico and 71 from Brazil. ChAdOx1-S (*n* = 518, 45.8%) was the most frequent vaccine, followed by CoronaVac (*n* = 429, 37.9%), BNT162b2 (*n* = 159, 14.0%), mRNA-1273 (*n* = 17, 1.5%), Ad26.COV2 (*n* = 5, 0.4%), and Ad5-nCoV (*n* = 4, 0.4%). Fifty-two percent were men, and the mean (SD) age was 56 (16.1), where the eldest group was ChAdOx1-S [68 (12.0)] and the youngest, mRNA-1273 [31 (21.3)] (*p* < 0.001). Regarding the participants’ medical history, the most common comorbidities were obesity in 383 (33.9%) participants, hypertension in 356 (31.5%), and diabetes mellitus in 219 (19.4%). Table 1 shows the participants’ medical history.

### 3.1. SARS-CoV-2 Spike 1–2 IgG Antibodies during Six-Month Follow-Up

We analyzed the SARS-CoV-2 Spike 1–2 IgG antibody titers in the subjects according to the vaccine they received (vaccine group) and SARS-CoV-2 infection history (COVID-19 history). We divided infection history into three groups: (1) negative history, group 1 (dynamically decreasing through follow-up), (2) positive history before vaccination, group 2, and new cases through follow-up, group 3 (breakthrough infections). Table 2 shows the median S1/S2 IgG by vaccine type and time-point. Before vaccination, in subjects with a negative COVID-19 history, the median (IQR) of antibody titers was 3.8 (0) AU/mL, while in previously exposed patients, it was 93.6 (172.8) AU/mL.

Following the first dose, at around 21–28 days from vaccination, mRNA-1273 and BNT162b2 had the highest median antibody titers in group 1 in comparison with other vaccines [163.5 (5426.2) and 89.4 (97.1) AU/mL, respectively (*p* < 0.001)] and group 2 [4970 (14,072.5) and 3295 (3725.0) AU/mL, (*p* < 0.001)]. Regarding group 3, mRNA-1273 and ChAdOx1-S had no infections at this time point. CoronaVac had two cases and BNT162b2 had one. The median antibody levels were 211 (-) AU/mL. 

Between 21 and 28 days after the completion of the vaccination scheme, in group 1, mRNA-1273 [2370 (3300) AU/mL] and BNT162b2 [1080 (1855.5) AU/mL]) had the highest antibody levels. In group 2, mRNA-1273 [5585 (6580.0) AU/mL] and Ad5-vCoV [3915 (-) AU/mL]) induced the highest antibody titers (*p* < 0.001). Furthermore, in group 3, ChAdOx1-S [2620 (-) AU/mL] and BTN162b2 [1960 (-) AU/mL] had the highest antibody titers. The groups vaccinated with mRNA-1273, Ad5-vCoV, and Ad26.COV2.S had no new cases.

Six months after vaccination in group 1, there was a decline in the SARS-CoV-2 Spike 1–2 IgG antibody titers by 69.5%. At this time point, mRNA-1273 [336.0 (827.0) AU/mL] maintained the most elevated titers, followed by BNT162b2 [271.0 (649.7) AU/mL] (*p* < 0.001). In group 2, the antibody levels declined by 36.4%. BNT162b2 [948.5 (1793.9) AU/mL and Ad26.COV2.S [1330 (-) AU/mL] had the highest antibody titers in comparison with other vaccines, *p* < 0.001. In Group 3, patients that had a new infection after completing the scheme until the end of the follow-up had a median of 1850 (1596.0) for BNT162b2 AU/mL and 1305.5 (-) AU/mL for mRNA-1273 presenting the highest antibody titers; however, there was no significant difference with other vaccines (*p* = 0.082).

The SARS-CoV-2 Spike 1–2 IgG antibodies change with each shot applied and over time. Fluctuation of antibody titers over time was significant because the highest antibody titers were reached 21–28 days after the completion of the scheme for all vaccine groups, dropping six months after vaccination, regardless of SARS-CoV-2 infection history (*p* < 0.001). Figure 1 shows the antibody response of the most frequent SARS-CoV-2 vaccines.

### 3.2. Ordinary Least Square Model for the Antibody Titers at the 6-Month Follow-Up

A total of 975 subjects were included in the ordinary least square model to predict the antibody titers at the 6-month follow-up, of whom 462 (47.4%) subjects had been vaccinated with ChAdOx1-SARS-CoV-2, 366 (37.5%) with CoronaVac, 138 (14.2%) with BNT162b2, and 9 (0.9%) with mRNA-1273. The model reached an adjusted R-squared value of 0.541. The variables that correlated with higher antibody titers at the 6-month follow-up were: any previous SARS-CoV-2 infection (0.24, 95% CI 0.11–0.37, *p* < 0.001), any SARS-CoV-2 infection after the 2nd dose (1.17, 95% CI 1.01–1.33, *p* < 0.001), the antibody titers after 2nd dose for the vaccination with ChAdOx1-SARS-CoV-2 (0.67, 95% CI 0.40–0.95, *p* < 0.001) or with CoronaVac (0.48, 95% CI 0.19–0.76, *p* = 0.001), compared to BNT162b2. The variables that correlated with lower antibody titers at the 6-month follow-up were: the body mass index [−0.07, 95% CI (−0.13, −0.02), *p* = 0.012], being vaccinated with ChAdOx1-SARS-CoV-2 (−0.56, 95% CI (−0.85, −0.26), *p* < 0.001) or CoronaVac [−0.45, 95% CI (−0.74, −0.16), *p* = 0.002], compared to BNT162b2 (Table 3).

### 3.3. SARS-CoV-2 Infection 

Before and through the follow-up of the patients, the history of SARS-CoV-2 infection was reported. Before vaccination, there were 247 (21.8%) cases, of which subjects from the groups who received Ad26.COV2.S [3 (60.0%)] and Ad5-vCoV [2 (50.0%)] presented the majority of cases proportionally. The most frequent symptoms were headache [128 (57.4%)], myalgias [124 (55.6%)], and anosmia [112 (50.2%)]. Out of the 247 patients with infection before vaccination, 219 (91.3%) received ambulatory treatment, 19 (7.9%) were hospitalized, and 2 (0.8%) were admitted to the Intensive Care Unit (ICU). 

In the period between the first and second dose, four COVID-19 cases were reported, of which CoronaVac reported two cases, while ChAdOx1-S and BNT162b2 just one case each. The most frequent symptom was headache [3 (75.0%)]. Three out of the four patients (75.0%) received ambulatory treatment; however, one patient vaccinated with CoronaVac was hospitalized. None of the infected patients in this period needed supplementary oxygen.

After the second dose or complete one-dose scheme, until 210 days of follow-up, 183 (16.2%) patients were infected, and the vaccine with the most reported cases was CoronaVac [120 (28.0%)], followed by Ad5-vCoV [1 (25.0%)] and Ad26.COV2.S [1 (20.0%)]; see Table 4. Subjects reported cough [109 (61.2%)], odynophagia [92 (51.7%)], and headache [79 (44.4%)] as the most frequent symptoms. The frequency of symptoms varied according to the administered vaccine. Tiredness [11 (45.8%)] was a recurring symptom reported by recipients of BNT162b2, with myalgia in ChAdOx1-S [17 (50.0%)] and CoronaVac [52 (44.8)], and rhinorrhea in mRNA-1273 [2 (100%)] and ChAdOx1-S [15 (44.1%)]. The majority of the infected participants were treated at home [176 (98.9%)]. Only two patients, recipients of ChAdOx1-S and CoronaVac, were hospitalized, and out of the two, only the patient vaccinated with ChAdOx1-S needed supplementary oxygen administered by nasal cannula, see Table 4.

In this study, patients were exposed to five waves of COVID-19. In the first wave, they were exposed to the Original strain; in the second wave to Alpha, Beta, Gamma, Kappa, Lambda, Eta, and Epsilon; in the third wave to Delta variant; in the fourth to Omicron B.1; and in the fifth wave to Omicron B.4/B.5. In the first wave 121 (10.4%), in the second wave 115 (9.9%); in the third wave 67 (5.8%), in the fourth wave 90 (7.8%) and finally in the fifth wave only 4 (0.3%) subjects had symptomatic COVID-19 (Table 4).

### 3.4. Survival Analysis and Cox Proportional Hazard Model

A total of 966 subjects were included in the adjusted Cox proportional hazard model, including 462 (47.8%) persons vaccinated with ChAdOx1-SARS-CoV-2, 366 (37.9%) persons vaccinated with CoronaVac, and 138 (14.3%) persons vaccinated with BNT162b2. The variables associated with a lower risk of infection at any time after the 2nd vaccine dose were older age (HR = 0.77, 95% CI 0.60–0.99, *p* = 0.04), any previous COVID infections (HR = 0.45, 95% CI 0.27–0.74, *p* = 0.002), and the interaction term between the vaccine type CoronaVac and the presence of important comorbidities (HR = 0.33, 95% CI 0.13–0.81, *p* = 0.02). The variables associated with a greater risk of infection at any time after the 2nd vaccine dose were the presence of important comorbidities (HR = 3.51, 95% CI 1.66–7.43, *p* = 0.001) and CoronaVac compared to ChAdOx1-SARS-CoV-2 (HR = 1.93, 95% CI 0.99–3.74, *p* = 0.05). No significant difference was found between BNT162b2 and ChAdOx1-SARS-CoV-2 in the absence (*p* = 0.21) and the presence (*p* = 0.15) of important comorbidities. The assumptions of the model were verified. The complete results are shown in Table 5. Figure 2 shows the survival curves.

## 4. Discussion

This multicentric study compared the humoral response induced by vaccination, the infection symptoms, treatment, and predictors of infection in recipients of BNT162b2, mRNA-1273, ChAdOx1-S, CoronaVac, Ad26.COV2, and Ad5-nCoV throughout 210 days of follow-up. This study evaluated the efficacy of vaccination through different waves in which patients could be exposed to different variants. Even though SARS-CoV-2 Spike 1–2 IgG antibodies decreased over time, the infection was lower through time and variants in comparison with the baseline. 

In all vaccine groups, SARS-CoV-2 Spike 1–2 IgG antibodies increased after the first to the second vaccination dose. mRNA-1273 and BNT162b2 developed the highest immune responses of all vaccines, in line with previous studies [14,15,16,17]. 

Over time SARS-CoV-2 Spike 1–2 IgG antibody titers significantly dropped six months after the completion of the vaccination scheme. This waning in antibody titers was markedly observed in patients with negative SARS-CoV-2 infection history. Similar results were found in a study conducted by Glöckner et al. where SARS-CoV-2 IgG levels were analyzed in recipients of BNT162b2, mRNA-1273, and ChAdOx1-S and they concluded that, despite the type of vaccine, the IgG levels elicited by vaccination waned after six months [18]. A previous systematic review over twenty-seven studies that included BNT16b2, CHADOX1, Ad25.COV2.S, and mRNA-1273 showed that vaccine-induced protection builds rapidly after the first dose and peaks within 4–42 days after the second dose, before waning begins typically from 3 to 24 weeks, having a varied response related to immune responses and demographics [19]. An additional research article by Cambim Fonseca MH showed the same decrease in antibodies in patients vaccinated with CoronaVAC after six months of follow-up [20]. In our study, using the same methodology to follow immune responses, we found that all vaccines induced a peak at one month, and the levels were consistently lower at six months after completion of the dosing scheme.

The ordinary least square model for the antibody titers at the 6-month follow-up showed that any previous SARS-CoV-2 infection and any SARS-CoV-2 infection after the 2nd dose were associated with higher antibody titers. This finding confirms the consistency of our data and previous observations about a certain degree of natural immunity. Ali et al. and Levi et al. reported that vaccinated subjects with previous SARS-CoV-2 infection elicited higher SARS-CoV-2 IgG and neutralizing antibodies than those without previous exposure. Ali et al. also showed a faster decline of antibody titers in patients without a previous infection [12,21]. Additionally, in our OLS model, the coefficient for COVID infections after the 2nd dose was significantly greater than that for previous COVID infections. This is consistent with a waning of the antibody levels over time: more time from the last infection corresponds to fewer antibodies circulating in the bloodstream. Additionally, we found that the effect of BMI was minor, which probably reflects the comorbidities that persons with a high BMI in our study population have, which are linked to a worse immune response and antibody maintenance over time. Previous studies have reported a negative correlation between SARS-CoV-2 antibodies in patients infected with COVID-19 and lower BMI [22]. Regarding ChAdOx1-S and CoronaVac, it is interesting to note that they produced a lower baseline level of antibody titers at six months compared to BNT162b2 (negative coefficients for their isolated contribution). Still, if they produced a good response after the 2nd dose, this was more easily maintained over time (positive coefficients for the interaction terms). We may speculate that BNT162b2 gives, on average, a good response after the 2nd dose, which is quite well maintained over time regardless of the interpersonal variability, whereas ChAdOx1-S and CoronaVac may or may not give a good response after the 2nd dose; in the case where there is a good increase in the antibodies after the 2nd dose, then these are more easily maintained over time, in particular at the 6-month follow-up.

Zeng et al. conducted a meta-analysis where they found that every studied vaccine was effective against SARS-CoV-2 variants; however, mRNA vaccines appeared to have a higher effectiveness than non-mRNA vaccines [23]. As mentioned before, mRNA vaccines produced higher and more sustained antibody titers after vaccination; however, it was of interest to see if they could also protect from infection.

A previous small study that compared protection from different variants in subjects that received ChAdOx1-S, BNT162b2, and CoronaVac showed greater safety and longer antibody blocking activity of the first two in subjects that had not previously been infected with SARS-CoV-2. CoronaVac was only effective in previously infected subjects. In the same study, the uninfected ChAdOx1 nCoV-19 vaccinee sera demonstrated effective neutralizing antibody reactivity against naïve, Delta, Epsilon, Alpha, Gamma, and Beta. BNT162b2 uninfected vaccinee sera showed an overall similar neutralizing antibody response as the ChAdOx1 nCoV-19 uninfected vaccinee sera, except that BNT162b2 sera provided better protection against Epsilon, Beta, and Gamma variants. In contrast, the CoronaVac uninfected vaccinee sera showed an effective response toward naïve and Delta only with the loss of activity seen against the Alpha, Epsilon, Beta, and Gamma variants (38%). All the uninfected vaccinee sera failed to neutralize the Omicron variant, while the sera of previously infected subjects showed neutralizing activity only for BNT162b2 and ChAdOx1 and not for CoronaVAC [24]. Our study showed that the infection rate was reduced by all vaccines, given its decrease from 21.8% (before vaccination) to 16.2% (after the completion of the scheme), despite the behavior and mutability of the virus that led to new waves of infections (from original to Omicron). Over the 6-month follow-up, more subjects vaccinated with CoronaVac were infected with SARS-CoV-2, while subjects vaccinated with ChAdOx1-S had the smallest number of infections. However, BNT162b2 had the highest decrease in the infection rate, declining by 14.5%, compared to ChAdOx1-S with just 5.7%. These results show that vaccination may not be effective enough to enhance the resistance against all virus exposures. Additionally, it is important to note that there was an antibody decrease over time, supporting the need of a booster dose and the development of new vaccines that protect against new variants [25,26]. Finally, and in agreement with observations elsewhere, further vaccination may not prevent clinical cases but associates in general with decrease in number of severe cases [6].

Additionally, our Cox regression model showed that individuals with a previous SARS-CoV-2 infection appeared to be at lower risk of infection after the 2nd dose at any time during the follow-up than those without any previous SARS-CoV-2 infection. This has been confirmed elsewhere [27]. The protective effect of older age may be due to a greater risk of exposure of younger persons than older ones, but we do not have data to support this speculation. The combined effect of vaccine type, comorbidities, and interaction terms may be difficult to interpret. We can summarize the impact of these variables as follows: in the absence of important comorbidities, setting the hazard ratio of ChAdOx1-S to 1 resulted in BNT162b2 having the same hazard ratio as ChAdOx1-S (*p* = 0.21), while the hazard ratio of CoronaVac was 1.93 (*p* = 0.05), thus making ChAdOx1-S and BNT162b2 a better choice for people without relevant comorbidities; in the presence of relevant comorbidities (i.e., diabetes, arthritis, or dyslipidemia), the hazard ratio for BNT162b2 and ChAdOx1-S were the same and higher than in the absence of comorbidities (HR = 3.51). However, the hazard ratio for CoronaVac was lower (HR = 2.25), thus making it a better choice compared to the other two vaccines in the presence of relevant comorbidities. These real-life findings may help define the choice of vaccines by public health systems.

Throughout the follow-up, the presentation of symptoms related to the SARS-CoV-2 infection changed. We noticed that the most frequent symptoms before vaccination were systemic symptoms, such as headache, myalgias, and anosmia. However, after completion of the vaccination scheme, symptoms shifted to mainly upper respiratory symptoms, like cough and odynophagia. Multiple factors are involved in changing the symptoms associated with SARS-CoV-2 infection. Mun-Keat stated that the change in the prevalence of symptoms is due to vaccination, immunity developed from previous SARS-CoV-2 infections, and the evolution and surge of new variants. 

The novelty and implications of our study rely on the fact that we compared the efficacy through seroconversion and infection rate of six different vaccines, showing a positive effect in all of them as all subjects showed an antibody increase after the completion of the scheme and were protected differently according to the vaccine received against SARS-CoV-2 infection through different waves. mRNA-based vaccines showed higher protection and a higher level of antibodies after the second dose. The antibodies decreased in all vaccines after six months of follow-up; predictors for the change were BMI, previous COVID-19 infection before and after vaccination, and the immunization with CoronaVac. The highest predictors for infection were age, vaccination with CoronaVaC, and the presence of comorbidities such as diabetes, rheumatoid arthritis, and dyslipidemia. Of note, from our Cox proportional hazard model, CoronaVaC appeared slightly more effective than the RNA vaccines in individuals with diabetes, rheumatoid arthritis, or dyslipidemia. The infection rate did not correlate with the level of antibodies reached after the complete scheme. Considering all this, we believe that vaccination should be encouraged in all countries, ages, and health conditions with the vaccine type that is accessible. However, we also must consider that new types of vaccines that cover new variants are mandatory for the future. 

These findings need to be taken with caution. One of the limitations of this study is the many correlations that exist across the various categories (e.g., sex, age, vaccine type, comorbidities), which make it difficult to precisely disentangle every single contribution. We considered that people started the vaccination schedule and follow-up period at different times, possibly including or excluding the appearance of important COVID-19 waves in their respective countries. Thus, in our model, we stratified by the follow-up date (whether before or after 15 December 2021, considered as the beginning of the important COVID wave in Mexico to which many persons in the study could or could not be exposed), but this may not be sufficient. In some vaccinated groups, there was a small sample size. Therefore, we only provided descriptive statistics in these cases, and did not include the data in the OLS and Cox proportional hazard model. Studies with larger sample sizes for these specific vaccines with a small sample size should be conducted in the future. Boosts with different vaccine types have been implemented all over the world. Studies using boosts with heterologous combinations in a real-world setting must be encouraged.

## 5. Conclusions

This multicentric study compared the humoral response induced by vaccination, the infection rate, symptoms, treatment, and predictors of infection in recipients of BNT162b2, mRNA-1273, ChAdOx1-S, CoronaVac, Ad26.COV2, and Ad5-nCoV throughout 210 days of follow-up. This study showed positive antibody responses in all vaccinated subjects with a decrease in antibody levels over six (+/− one) months of follow-up. The variables that correlated with higher antibody titers were any previous SARS-CoV-2 infection, any SARS-CoV-2 infection after the 2nd dose, and the antibody titers after 2nd dose for a vaccination with ChAdOx1-SARS-CoV-2 and with CoronaVac, compared to BNT162b2. 

If we compare the infection rate prior to vaccination and during follow-up, the latter was lower through all waves and variants. However, the predictors of infection were the vaccination with CoronaVac in comparison to BNT162b2 and ChAdOx1-S. In the presence of comorbidities such as diabetes, rheumatoid arthritis, or dyslipidemia, CoronaVac lowered the risk of infection.

## Figures and Tables

**Figure 1 vaccines-11-00842-f001:**
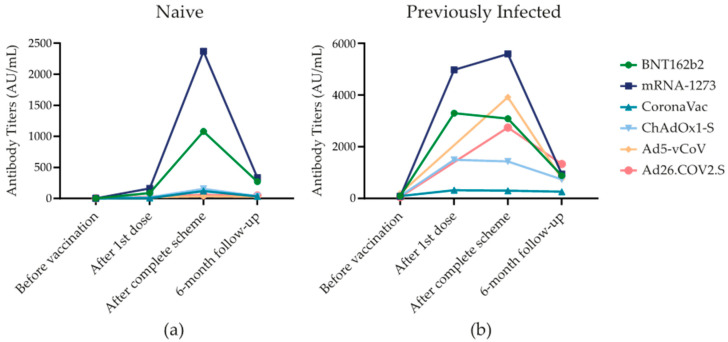
SARS-CoV-2 Spike 1–2 IgG antibodies over six-month follow-up. (**a**) SARS-CoV-2 Spike 1–2 IgG antibody levels (AU/mL) in subjects that were not infected with SARS-CoV-2 that were exposed to one of the six different types of vaccines. (**b**) SARS-CoV-2 Spike 1–2 IgG antibody levels (AU/mL) in subjects that were previously infected with SARS-CoV-2 before vaccination and that were exposed to one of the six different types of vaccines.

**Figure 2 vaccines-11-00842-f002:**
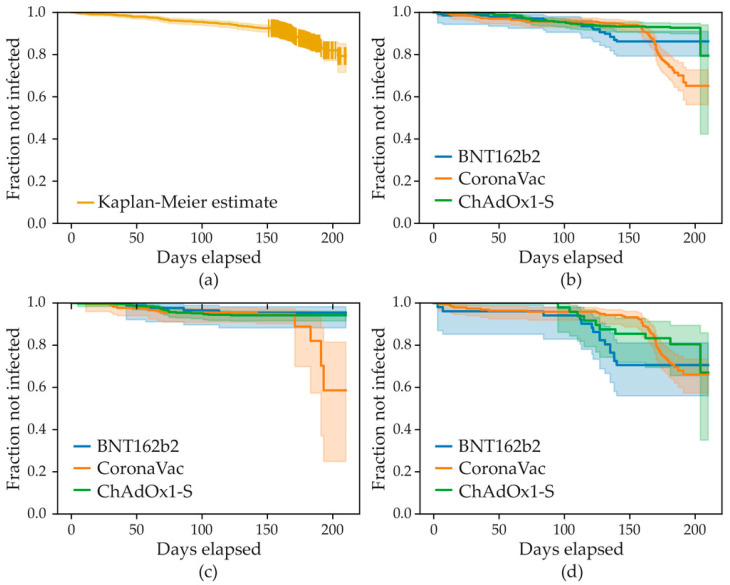
Survival analysis curves over 210 days follow-up on the x axis. (**a**) Kaplan–Meier estimate for the whole population. (**b**) Kaplan–Meier estimates stratified by vaccine type: BNT162b2, ChAdOx1-S, and CoronaVac. (**c**) Kaplan–Meier estimates stratified by vaccine and date: before 15 December 2021 for BNT162b2, ChAdOx1-S, and CoronaVac vaccines. (**d**) Kaplan–Meier estimates stratified by vaccine and date: after 15 December 2021 for BNT162b2, ChAdOx1-S, and CoronaVac vaccines.

**Table 1 vaccines-11-00842-t001:** Medical history.

	Total (*n* = 1132) (%)	BNT162b2 (*n* = 159) (%)	mRNA-1273 (*n* = 17) (%)	CoronaVac (*n* = 429) (%)	ChAdOx1-S (*n* = 518) (%)	Ad5-vCoV (*n* = 4) (%)	Ad26. COV2.S (*n* = 5) (%)	*p*-Value
Diabetes Mellitus 2	219 (19.4)	6 (3.8)	1 (5.9)	63 (14.7)	149 (28.8)	0 (0)	0 (0)	<0.001
Hypertension	356 (31.5)	21 (13.3)	2 (11.8)	94 (22.0)	237 (45.8)	2 (50.0)	0 (0)	<0.001
Asthma	32 (2.8)	3 (1.9)	1 (5.9)	13 (3.0)	15 (2.9)	0 (0)	0 (0)	0.923
Chronic Obstructive Pulmonary Disease	8 (0.7)	0 (0)	0 (0)	2 (0.5)	6 (1.2)	0 (0)	0 (0)	0.675
Obesity	383 (33.9)	33 (20.9)	2 (12.5)	142 (33.2)	205 (39.6)	1 (25.0)	0 (0)	<0.001
Smoking	99 (8.5)	17 (10.8)	4 (23.5)	43 (10.0)	34 (6.6)	0 (0)	1 (20.0)	0.060
Renal Disease	15 (1.3)	1 (0.6)	0 (0)	4 (0.9)	10 (1.9)	0 (0)	0 (0)	0.719
Dyslipidemia	210 (18.6)	12 (7.6)	1 (5.9)	59 (13.8)	137 (26.4)	1 (25.0)	0 (0)	<0.001
Pregnancy	2 (0.4)	0 (0)	0 (0)	1 (0.2)	0 (0)	1 (25.0)	0 (0)	<0.001
Active Neoplasia	12 (1.1)	0 (0)	0 (0)	1 (0.2)	11 (2.1)	0 (0)	0 (0)	0.067
Previous Neoplasia	32 (2.8)	1 (0.6)	0 (0)	6 (1.4)	25 (4.8)	0 (0)	0 (0)	0.015
Atrial Fibrillation	31 (2.7)	0 (0)	0 (0)	4 (0.9)	27 (5.2)	0 (0)	0 (0)	<0.001
Cardiac Failure	9 (0.8)	0 (0)	0 (0)	0 (0)	9 (1.7)	0 (0)	0 (0)	0.057
Coronary Disease	23 (2.0)	0 (0)	0 (0)	3 (0.7)	20 (3.9)	0 (0)	0 (0)	0.006
Stroke	14 (1.2)	0 (0)	0 (0)	1 (0.2)	13 (2.5)	0 (0)	0 (0)	0.027
Hepatic Steatosis	50 (4.4)	3 (1.9)	0 (0)	23 (5.4)	23 (4.4)	1 (25.0)	0 (0)	0.140
Cirrhosis	6 (0.5)	0 (0)	0 (0)	1 (0.2)	5 (1.0)	0 (0)	0 (0)	0.616
Hepatitis/Hepatic Failure	13 (1.2)	1 (0.6)	0 (0)	4 (0.9)	8 (1.5)	0 (0)	0 (0)	0.906
Rheumatoid Arthritis	58 (5.1)	5 (3.2)	1 (5.9)	10 (2.3)	42 (8.1)	0 (0)	0 (0)	0.003
Psoriasis/Thyroiditis and Other Immune Diseases	77 (6.8)	6 (3.8)	0 (0)	21 (4.9)	50 (9.7)	0 (0)	0 (0)	0.022
Usage of Immunosuppressive Drugs	16 (1.4)	2 (1.3)	0 (0)	5 (1.2)	9 (1.7)	0 (0)	0 (0)	0.965
Gout	39 (3.5)	2 (1.3)	0 (0)	11 (2.6)	26 (5.0)	0 (0)	0 (0)	0.158
Surgical Procedure with General Anesthesia	64 (5.7)	4 (2.5)	0 (0)	19 (4.4)	39 (7.5)	2 (50.0)	0 (0)	<0.001
Organ Transplant	5 (0.4)	0 (0)	0 (0)	1 (0.2)	4 (0.8)	0 (0)	0 (0)	0.773

Data are presented as frequencies and percentages. Chi square test was used for comparison. A *p*-value < 0.05 was considered statistically significant.

**Table 2 vaccines-11-00842-t002:** Median IgG SARS-CoV-2 S1-S2 antibody titers by vaccine and SARS-CoV-2 infection history.

SARS-CoV-2 Infection History	Total (*n* = 1132)	BNT162b2 (*n* = 159) (IQR) AU/mL	mRNA-1273 (*n* = 17) (IQR) AU/mL	CoronaVac (*n* = 429) (IQR) AU/mL	ChAdOx1-S (*n* = 518) (IQR) AU/mL	Ad5-vCoV (*n* = 4) (IQR) AU/mL	Ad26.COV2.S (*n* = 5) (IQR) AU/mL	*p*-Value
Before vaccination								
Negative	3.8 (0) (*n* = 884)	3.8 (3.7) (*n* = 110)	3.8 (29.1) (*n* = 13)	3.8 (1.2) (*n* = 303)	3.8 (0) (*n* = 454)	3.8 (0) (*n* = 2)	3.8 (0) (*n* = 2)	0.009
Positive	96.8 (173.7) (*n* = 247)	99.6 (178.5) (*n* = 48)	77.5 (196.6) (*n* = 4)	91.0 (137.0) (*n* = 126)	90.7 (267.0) (*n* = 64)	198.0 (-) (*n* = 2)	56.9 (-) (*n* = 3)	0.908
*p*-value	<0.001	<0.001	0.032	<0.001	<0.001	0.333	0.200	
After first dose								
Negative	20.3 (69.6) (*n* = 866)	89.4 (97.1) (*n* = 107)	163.5 (5426.2) (*n* = 12)	6.56 (22.1) (*n* = 249)	21.6 (45.9) (*n* = 453)	-	-	<0.001
Positive before vaccination	584.0 (2409.0) (*n* = 235)	3295 (3725.0) (*n* = 48)	4970 (14,072.5) (*n* = 4)	320.0 (447.0) (*n* = 119)	1495.0 (3141.7) (*n* = 64)	-	-	<0.001
New cases	211 (-) (*n* = 3)	211 (-) (*n* = 1)	(*n* = 0)	245 (-) (*n* = 2)	(*n* = 0)	-	-	1.000
*p*-value	<0.001	<0.001	0.078	<0.001	<0.001	-	-	
21–28 days after completion of scheme								
Negative	160 (259.2) (*n* = 803)	1080 (1855.5) (*n* = 94)	2370 (3300) (*n* = 11)	119 (112.9) (*n* = 269)	155.0 (212.8) (*n* = 425)	37.25 (-) (*n* = 2)	60.7 (-) (*n* = 2)	<0.001
Positive before vaccination	608.5 (2156.0) (*n* = 228)	3085.0 (3292.5) (*n* = 42)	5585.0 (6580.0) (*n* = 4)	303.0 (376.5) (*n* = 117)	1430.0 (3174.7) (*n* = 60)	3915 (-) (*n* = 2)	2740 (-) (*n* = 3)	<0.001
New cases	1555 (1342.7) (*n* = 12)	1960 (-) (*n* = 3)	(*n* = 0)	905.5 (1363.5) (*n* = 8)	2620.0 (-) (*n* = 1)	(*n* = 0)	(*n* = 0)	0.081
*p*-value	<0.001	<0.001	0.056	<0.001	<0.001	0.333	0.200	
Six months after completion of scheme								
Negative	48.8 (152.2) (*n* = 730)	271.0 (649.7) (*n* = 92)	336.0 (827.0) (*n* = 11)	33.7 (131.4) (*n* = 203)	38.5 (72.9) (*n* = 421)	27.9 (-) (*n* = 1)	49.2 (-) (*n* = 2)	<0.001
Positive before vaccination	387.0 (834.0) (*n* = 247)	886.5 (1079.0) (*n* = 48)	948.5 (1793.9) (*n* = 4)	262 (441.8) (*n* = 126)	736.0 (1328.5) (*n* = 64)	765 (-) (*n* = 2)	1330.0 (-) (*n* = 3)	<0.001
New cases	926 (2353.0) (*n* = 155)	1850 (1596.0) (*n* = 19)	1305.5 (-) (*n* = 2)	372.5 (2886.6) (*n* = 100)	1190 (1901.5) (*n* = 33)	3.8 (-) (*n* = 1)	(*n* = 0)	0.082
*p*-value	<0.001	<0.001	0.599	<0.001	<0.001	0.259	0.200	
*p*-value	<0.001	<0.001	<0.001	<0.001	<0.001	0.022	0.015	

Data are presented as median and interquartile ranges. Mann–Whitney U, Kruskal–Wallis, and Friedman tests were used for comparison. A *p*-value less than 0.05 was considered significant.

**Table 3 vaccines-11-00842-t003:** Ordinary least square model for the antibody titers at 6-month follow-up ^a^.

	Coefficient	*p*-Value
Intercept	0.21 (−0.06, 0.49)	0.13
ChAdOx1-SARS-CoV-2	−0.56 (−0.85, −0.26)	<0.001
CoronaVac	−0.45 (−0.74, −0.16)	0.002
mRNA−1273	−0.52 (−1.8, 0.77)	0.43
Male sex	0.03 (−0.07, 0.13)	0.58
Age ^b^	−0.04 (−0.12, 0.03)	0.25
BMI ^c^	−0.07 (−0.13, −0.02)	0.012
Antibodies ^d^	0.12 (−0.15, 0.38)	0.39
Antibodies*ChAdOx1-SARS-CoV-2 ^d^	0.67 (0.4, 0.95)	<0.001
Antibodies*CoronaVac ^d^	0.48 (0.19, 0.76)	0.001
Antibodies*mRNA-1273 ^d^	0.29 (−0.81, 1.39)	0.60
Previous COVID infection ^e^	0.24 (0.11, 0.37)	<0.001
COVID infection after 2nd dose	1.17 (1.01, 1.33)	<0.001

^a^ Abbreviations: BMI, body mass index. Reference: female vaccinated with BNT162b2, without previous COVID infections nor COVID infections after the 2nd dose. The target variable, antibody titers at 6-month follow-up in AU/mL, was standardized (mean = 0, sd = 1), SD = 1664. ^b^ In years, standardized (mean = 0, sd = 1), SD = 15.6. ^c^ In kg/m^2^, standardized (mean = 0, sd = 1), SD = 5.2. ^d^ Antibody titers in AU/mL were standardized (mean = 0, sd = 1), SD = 2208. ^e^ Before the first dose. * is interaction term.

**Table 4 vaccines-11-00842-t004:** SARS-CoV-2 infection history.

SARS-CoV-2 Infection	Total (*n* = 1132) (%)	BNT162b2 (*n* = 159) (%)	mRNA-1273 (*n* = 17) (%)	CoronaVac (*n* = 429) (%)	ChAdOx1 (*n* = 518) (%)	Ad5-vCoV (*n* = 4) (%)	Ad26.COV2.S (*n* = 5) (%)	*p*-Value
Before vaccination	247 (21.8)	48 (30.4)	4 (23.5)	126 (29.4)	64 (12.4)	2 (50.0)	3 (60.0)	<0.001
After first dose	4 (0.4)	1 (0.6)	0 (0)	2 (0.5)	1 (0.2)	-	-	0.818
After second dose	183 (16.2)	25 (15.7)	2 (11.8)	120 (28.0)	34 (6.6)	1 (25.0)	1 (20.0)	<0.001
SARS-CoV-2 Infection before vaccination								
Symptoms								
Fever	87 (39.0)	17 (40.5)	3 (75.0)	46 (41.1)	18 (30.0)	2 (100)	1 (33.3)	0.178
Feverish	55 (24.7)	10 (23.8)	2 (50.0)	33 (29.5)	10 (16.7)	0 (0)	0 (0)	0.261
Cough	111 (49.8)	20 (47.6)	3 (75.0)	56 (50.0)	29 (48.3)	1 (50.0)	2 (66.7)	0.914
Headache	128 (57.4)	22 (52.4)	2 (50.0)	69 (61.6)	32 (53.3)	1 (50.0)	2 (66.7)	0.864
Dyspnea	70 (31.4)	13 (31.0)	0 (0)	34 (30.4)	22 (36.7)	1 (50.0)	0 (0)	0.499
Conjunctivitis	9 (4.0)	1 (2.4)	0 (0)	5 (4.5)	3 (5.0)	0 (0)	0 (0)	0.972
Palpitations	29 (13.0)	8 (19.0)	1 (25.0)	11 (9.8)	9 (15.0)	0 (0)	0 (0)	0.575
Thoracic pain	47 (21.1)	10 (23.8)	0 (0)	28 (25.0)	8 (13.3)	1 (50.0)	0 (0)	0.281
Odynophagia	81 (36.3)	17 (40.5)	2 (50.0)	37 (33.0)	22 (36.7)	1 (50.0)	2 (66.7)	0.774
Myalgias	124 (55.6)	25 (59.5)	3 (75.0)	62 (55.4)	32 (53.3)	1 (50.0)	1 (33.3)	0.898
Arthralgias	84 (37.7)	15 (31.3)	2 (50.0)	43 (38.4)	22 (36.7)	1 (50.0)	1 (33.3)	0.991
Anosmia	112 (50.2)	20 (47.6)	1 (25.0)	59 (42.7)	29 (48.3)	1 (50.0)	2 (66.7)	0.874
Tiredness	61 (27.4)	14 (33.3)	2 (50.0)	23 (20.5)	21 (35.0)	1 (50.0)	0 (0)	0.166
Diarrhea	42 (18.8)	10 (23.8)	0 (0)	17 (15.2)	15 (25.0)	0 (0)	0 (0)	0.387
Vomiting	23 (10.3)	5 (11.9)	1 (25.0)	8 (7.1)	9 (15.0)	0 (0)	0 (0)	0.512
Nausea	11 (4.9)	1 (2.4)	0 (0)	2 (1.8)	8 (13.3)	0 (0)	0 (0)	0.029
Treatment								
Ambulatory	219 (91.3)	43 (91.5)	3 (75.0)	112 (93.3)	56 (87.5)	2 (100)	3 (100)	0.707
Hospitalization	19 (7.9)	4 (2.5)	1 (25.0)	6 (5)	8 (12.5)	0 (0)	0 (0)	
Intensive Care Unit	2 (0.8)	0 (0)	0 (0)	2 (1.7)	0 (0)	0 (0)	0 (0)	
Need of supplementary oxygen								0.936
Total	17 (7)	3 (6.4)	0 (0)	8 (6.6)	6 (9.4)	0 (0)	0 (0)	
Nasal cannula	11 (64.7)	2 (66.7)	0 (0)	4 (50.0)	5 (83.3)	0 (0)	0 (0)	
Non-rebreather mask	2 (11.8)	0 (0)	0 (0)	1 (12.5)	1 (16.7)	0 (0)	0 (0)	0.684
High flow equipment	3 (17.6)	1 (33.3)	0 (0)	2 (25.0)	0 (0)	0 (0)	0 (0)	
Orotracheal intubation	1 (5.9)	0 (0)	0 (0)	1 (12.5)	0 (0)	0 (0)	0 (0)	
SARS-CoV-2 Infection after first dose								
Symptoms								
Fever	2 (50.0)	1 (100)	0 (0)	1 (50.0)	0 (0)	-	-	0.368
Feverish	2 (50.0)	0 (0)	0 (0)	2 (100)	0 (0)	-	-	0.135
Cough	2 (50.0)	0 (0)	0 (0)	2 (100)	0 (0)	-	-	0.135
Headache	3 (75.0)	1 (100)	0 (0)	1 (50.0)	1 (100)	-	-	0.513
Dyspnea	1 (25.0)	1 (100)	0 (0)	0 (0)	0 (0)	-	-	0.135
Irritability	1 (25.0)	1 (100)	0 (0)	0 (0)	0 (0)	-	-	0.135
Palpitations	1 (25.0)	0 (0)	0 (0)	1 (50.0)	0 (0)	-	-	0.513
Chills	2 (50.0)	0 (0)	0 (0)	2 (100)	0 (0)	-	-	0.135
Odynophagia	2 (50.0)	1 (100)	0 (0)	1 (50.0)	0 (0)	-	-	0.368
Arthralgias	2 (50.0)	0 (0)	0 (0)	1 (50.0)	1 (100)	-	-	0.368
Anosmia	1 (25.0)	0 (0)	0 (0)	1 (50.0)	0 (0)	-	-	0.513
Tiredness	1 (25.0)	0 (0)	0 (0)	1 (50.0)	0 (0)	-	-	0.513
Diarrhea	1 (25.0)	0 (0)	0 (0)	1 (50.0)	0 (0)	-	-	0.513
Vomiting	1 (25.0)	0 (0)	0 (0)	1 (50.0)	0 (0)	-	-	0.513
Nausea	1 (25.0)	0 (0)	0 (0)	0 (0)	1 (100)	-	-	0.135
Dysgeusia	1 (25.0)	0 (0)	0 (0)	1 (50.0)	0 (0)	-	-	0.513
Rhinorrhea	1 (25.0)	1 (100)	0 (0)	0 (0)	0 (0)	-	-	0.135
Polypnea	1 (25.0)	1 (100)	0 (0)	0 (0)	0 (0)	-	-	0.135
Treatment								
Ambulatory	3 (75.0)	1 (100)	0 (0)	1 (50.0)	1 (100)	-	-	0.513
Hospitalization	1 (25.0)	0 (0)	0 (0)	1 (50.0)	0 (0)	-	-	
Intensive Care Unit	0 (0)	0 (0)	0 (0)	0 (0)	0 (0)	-	-	
Need of supplementary oxygen								
Total	0 (0)	0 (0)	0 (0)	0 (0)	0 (0)	-	-	-
SARS-CoV-2 Infection after complete vaccination scheme								
Symptoms								
Fever	54 (30.3)	8 (33.3)	1 (50)	37 (31.9)	8 (23.5)	0 (0)	0 (0)	0.818
Feverish	25 (14.0)	4 (16.7)	0 (0)	16 (13.8)	5 (14.7)	0 (0)	0 (0)	0.976
Cough	109 (61.2)	9 (37.5)	1 (50)	78 (67.2)	20 (58.8)	1 (100)	0 (0)	0.079
Headache	79 (44.4)	13 (54.2)	1 (50.0)	50 (43.1)	13 (38.2)	1 (100)	1 (100)	0.541
Dyspnea	11 (6.2)	1 (4.2)	0 (0)	6 (5.2)	4 (11.8)	0 (0)	0 (0)	0.782
Irritability	6 (3.4)	1 (4.2)	0 (0)	3 (2.6)	2 (5.9)	0 (0)	0 (0)	0.957
Conjunctivitis	8 (4.5)	0 (0)	0 (0)	3 (2.6)	4 (11.8)	1 (100)	0 (0)	<0.001
Palpitations	5 (2.8)	1 (4.2)	0 (0)	3 (2.6)	1 (2.9)	0 (0)	0 (0)	0.998
Thoracic pain	13 (7.3)	1 (4.2)	0 (0)	8 (6.9)	3 (8.8)	1 (100)	0 (0)	0.020
Chills	42 (23.6)	3 (12.5)	0 (0)	33 (28.4)	6 (17.6)	0 (0)	0 (0)	0.409
Odynophagia	92 (51.7)	14 (58.3)	2 (100)	60 (51.7)	14 (41.2)	1 (100)	1 (100)	0.340
Myalgias	77 (43.3)	7 (29.2)	1 (50)	52 (44.8)	17 (50.0)	0 (0)	0 (0)	0.514
Arthralgias	53 (29.8)	5 (20.8)	0 (0)	34 (29.3)	13 (38.2)	1 (100)	0 (0)	0.334
Anosmia	37 (20.8)	4 (16.7)	0 (0)	20 (17.2)	12 (35.3)	0 (0)	0 (0)	0.716
Tiredness	58 (32.6)	11 (45.8)	1 (50.0)	34 (29.3)	11 (32.4)	1 (100)	0 (0)	0.379
Diarrhea	16 (9.0)	2 (8.3)	0 (0)	9 (7.8)	5 (14.7)	0 (0)	0 (0)	0.852
Vomiting	3 (1.7)	0 (0)	0 (0)	2 (1.7)	1 (2.9)	0 (0)	0 (0)	0.977
Dysgeusia	22 (12.4)	1 (4.2)	0 (0)	13 (11.2)	7 (20.6)	0 (0)	1 (100)	0.046
Rhinorrhea	75 (42.1)	9 (37.5)	2 (100)	48 (41.4)	15 (44.1)	1 (100)	0 (0)	0.399
Polypnea	3 (1.7)	1 (4.2)	0 (0)	2 (1.7)	0 (0)	0 (0)	0 (0)	0.908
Abdominal pain	4 (2.2)	0 (0)	0 (0)	1 (0.9)	2 (5.9)	1 (100)	0 (0)	<0.001
Treatment								
Ambulatory	176 (98.9)	25 (100)	2 (100)	115 (99.1)	32 (97.0)	1 (100)	1 (100)	0.915
Hospitalization	2 (1.1)	0 (0)	0 (0)	1 (0.9)	1 (3.0)	0 (0)	0 (0)	
Intensive Care Unit	0 (0)	0 (0)	0 (0)	0 (0)	0 (0)	0 (0)	0 (0)	
Need of supplementary oxygen								
Total	0 (0)	0 (0)	0 (0)	0 (0)	1 (3.0)	0 (0)	0 (0)	0.495
Nasal cannula	1 (100)	0 (0)	0 (0)	0 (0)	1 (100)	0 (0)	0 (0)	-

Data are presented as frequencies and percentages. Chi square test was used for comparison. A *p*-value less than 0.05 was considered statistically significant.

**Table 5 vaccines-11-00842-t005:** Adjusted Cox Proportional Hazard Model.

	HR (95% CI)	*p*-Value
BMI ^a^	0.97 (0.86–1.09)	0.56
Previous COVID infections ^b^	0.45 (0.27–0.74)	0.002
Age ^c^	0.77 (0.60–0.99)	0.04
Male sex	1.19 (0.83–1.72)	0.35
Antibody level after second dose ^d^	1.01 (0.81–1.25)	0.94
BNT162b2	1.73 (0.73–4.08)	0.21
CoronaVac	1.93 (0.99–3.74)	0.05
Comorbidities ^e^	3.51 (1.66–7.43)	0.001
BNT162b2*comorbidities	0.30 (0.06–1.56)	0.15
CoronaVac*comorbidities	0.33 (0.13–0.81)	0.02

Abbreviations: HR, hazard ratio; BMI, body mass index. Reference: female with no previous COVID infections and no relevant comorbidities, vaccinated with ChAdOx1-SARS-CoV-2. The model was stratified by the follow-up date (before or after 15 December 2021). ^a^ In kg/m^2^, standardized (mean = 0, sd = 1) then squared. ^b^ Before the first vaccine dose. ^c^ In years, standardized (mean = 0, sd = 1). ^d^ In AU/mL, standardized (mean = 0, sd = 1). ^e^ At least one of: dyslipidemia, diabetes, or arthritis. * interaction term.

## Data Availability

Data are available upon reasonable request to corresponding author.

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
