# Peer review of "Efficacy of Six Different SARS-CoV-2 Vaccines during a Six-Month Follow-Up and Five COVID-19 Waves in Brazil and Mexico"

_vaccines, 2023, doi:10.3390/vaccines11040842_

Round 1
Reviewer 1 Report
The authors copmare six different SARS-CoV-2 vaccines in terms of seroconversion efficacy and infection rate.
The paper is very well written ans all the section are extremely detailed.
There is no particular novelty in the paper, but the amount of patients, the number of vaccines compared and the completeness of statistical analysis performed justify the need for publication.
There are no particular concerns according to this reviewer.
Author Response
Thank you for your comments
Reviewer 2 Report
The authors studied the efficacy of six different SARS-CoV-2 vaccines and concluded that these vaccines induce a humoral immune response significantly. However, there are some concerns that need to be addressed:
• The results suggest that natural immunity may be highly contributing to the elevated level of antibody response, and an appropriate negative control group is missing to demonstrate the actual vaccine-mediated effects. Without the appropriate control group, no concrete conclusion can be made from this study.
• The results suggest that vaccination may not be effective enough to enhance resistance against the virus. A substantial percentage of clinical diseases occurred post-vaccination, which is a serious problem of this mass vaccination and needs to be discussed in detail, comparing with other studies.
• In some groups, the sample size is not sufficient, which could affect the statistical power of the study.
• The implications of the findings and the limitations of this study need to be discussed in detail in the discussion section.
• The title does not reflect the main focus of the study, which seems to be mainly on the humoral immune response, rather than the vaccine efficacy. It would be better to revise the title to reflect the actual focus of the study.
Author Response
Thank you for taking the time to review our manuscript and making comments to improve it.
Find below how we addressed them, and you will find them in the manuscript highlighted in red.
- The results suggest that natural immunity may be highly contributing to the elevated level of antibody response, and an appropriate negative control group is missing to demonstrate the actual vaccine mediated effects. Without appropriate control group, no concrete conclusion can be made from this study
R= This is a real-life observational study of subjects that were vaccinated according to government indications and timelines. Since we were in a pandemic scenario and it is well described that vaccines are the best option for COVD-19 prevention, we considered appropriate in the design to measure a subject’s IgG baseline before vaccination as a control and the comparison change of IgG after complete vaccination scheme in different timepoints. The timepoint just after the vaccination scheme also contains the response due to vaccines only, without any infection. This allows to identify the vaccine mediated effect. It would clearly be unethical to include a control non- vaccinated group. In addition, our aim was to compare the response with different types of vaccine and not necessarily compare subjects with and without further vaccination – at the time this was not an option due to local governmental policies. We think that this is a good design for making conclusions since all subjects had a baseline IgG before vaccination.
- The results suggest that vaccination may not be effective enough to enhance resistance against the virus. A substantial percentage of clinical diseases occurred post vaccination, which is a serious problem of this mass vaccination and needs to be discussed in detail, comparing with other studies.
R= Our study showed that the infection rate was reduced by all vaccines, given its decrease from 21.8% (before vaccination) to 16.2% (after the complete scheme). Over the 6-month follow-up, more subjects vaccinated with CoronaVac got infected with SARS-CoV-2, while subjects vaccinated with ChAdOx1-S had the lowest infection rate. However, BNT162b2 had the highest decrease in the infection rate, declining 14.5%, compared to ChAdOx1-S, of just 5.7%. These results show that vaccination may not be effective enough to enhance resistance against all virus exposure. Additionally, it is important to note that there is an antibody decrease over time, supporting the need of a booster dose and the development of new vaccines that protect against new variants. Finally, and in agreement with observations elsewhere, further vaccination may not prevent clinical cases but associates in general with decrease in number of severe cases. We modified the manuscript to take this discussion in consideration and have added the appropriate references (Line 395-407).
In some groups the sample size is not sufficient, which could affect the statistical power.
R=Agree, for this reason we did not include small sample size groups in Ordinary Least square model and Cox regression.
The implications of the findings and the limitations of this study need to be discussed in detail in the discussion section.
Implications:
The novelty and implications of our study rely on the fact that we compared the efficacy through seroconversion and infection rate of six different vaccines, showing a positive effect in all of them as all subjects showed an antibody increase after the complete scheme and were protected differently according to the vaccine received against SARS-CoV-2 infection through different waves. mRNA-based vaccines showed higher protection and a higher level of antibodies after the second dose. The antibodies decreased in all vaccines after six months of follow-up; predictors for the change were BMI, previous COVID-19 infection before and after vaccination, and the immunization with CoronaVac. The highest predictors for infection were age, vaccination with CoronaVaC, and the presence of comorbidities such as diabetes, rheumatoid arthritis, and dyslipidemia. Of note, from our Cox proportional hazard model, CoronaVaC appeared slightly more effective than the RNA vaccines in individuals with diabetes, rheumatoid arthritis, or dyslipidemia. The infection rate did not correlate with the level of antibodies reached after the complete scheme. Considering all this, we believe that vaccination should be encouraged in all countries, ages, and health conditions with the vaccine type that is accessible. However, we must also consider that new types of vaccines that cover new variants are mandatory for the future. (Line 432-448)
R= We added this in the limitation of the study: One of the limitations of this study is the many correlations that exist across the various categories (e.g., sex, age, vaccine type, comorbidities), which make it difficult to precisely disentangle every single contribution. We considered that people started the vaccination schedule and follow-up period at different times, possibly including or excluding the appearance of important COVID-19 waves in their respective countries. Thus, in our model, we stratified by the follow-up date (whether before or after 15th December 2021, considered as the beginning of the important COVID wave in Mexico to which many persons in the study could or could not be exposed), but this may not be sufficient. Also, we had some vaccinated groups with small sample size, so we only included descriptive statistics about them, but they were not included in the OLS and Cox proportional hazard model. Studies with larger sample size in these specific small groups should be addressed in the future (Line 449-461).
The title does not reflect the main focus of the study, which seems to be mainly on the humoral immune response, rather than the vaccine efficacy. It would be better to revise the title to reflect the actual focus of the study
R= For this study Efficacy was measured through symptomatic infection and humoral response. We rephrased better this idea in the aim and method section to explain better the title. (line 8, 130-131)
Reviewer 3 Report
Dear authors,
Congratulations for your work. Your article is well structured and very interesting at the field.
My comments are:
1. Line 59: Use a fullstop instead of -.
2. Line 59: Add CoronaVac classification.
3. Line 91: Explain STROBE guidelines.
4. Table 1: Explain COPD.
5. Line 191: Delete word and.
6. Line 336: Use a fullstop instead of -.
7. Line 404: Determine the 2 vaccines.
8. In the abstract line 37 describe the "fourth" examination in another word.
Author Response
Thank you for taking your precious time to review our manuscript. Please find below how we addressed your comments. Also, you will find them highlighted in red in the manuscript.
- Line 59: Use a fullstop instead of -.
R= Corrected as suggested
- Line 59: Add CoronaVac classification.
R= Added in line 62 - 64. “Whole inactivated vaccines like CoronaVac (Sinovac Biotech) employ cultured inactivated viral particles containing antigens of the pathogen of interest able to induce immune response”
- Line 91: Explain STROBE guidelines.
R= Corrected as Suggested (lines 91-92) “This study was made following the Strengthening the Reporting of Observational studies in Epidemiology (STROBE) guidelines”
- Table 1: Explain COPD.
R= Corrected as suggested, Chronic Obstructive Pulmonary disease
- Line 191: Delete word and.
R= Corrected as suggested
- Line 336: Use a fullstop instead of -.
R= Corrected as suggested.
- Line 404: Determine the 2 vaccines.
R= Corrected as suggested.
- In the abstract line 37 describe the "fourth" examination in another word.
R= Corrected as suggested (line 36-37). “SARS-CoV-2 Spike 1-2 IgG levels were taken before receiving the first vaccine, 21 days after each dose, and the last sample at six months (+/-1 month) after the last dose.”
Round 2
Reviewer 2 Report
Thank you for addressing the comments and suggestions provided during the initial review process. The revisions made to the manuscript have significantly improved the clarity and structure of the manuscript. The authors have addressed most of the concerns raised during the initial review, and the manuscript now presents a clear and compelling argument. While there are still some minor issues that need to be addressed, I believe that the revised manuscript is now suitable for publication.